# Monocular Depth Estimation Using a Laplacian Image Pyramid with Local Planar Guidance Layers

**DOI:** 10.3390/s23020845

**Published:** 2023-01-11

**Authors:** Youn-Ho Choi, Seok-Cheol Kee

**Affiliations:** 1Department of Smart Car Engineering, Chungbuk National University, 1 Chungdae-ro, Seowon-gu, Cheongju-si 28644, Republic of Korea; 2Department of Intelligent Systems and Robotics, Chungbuk National University, 1 Chungdae-ro, Seowon-gu, Cheongju-si 28644, Republic of Korea

**Keywords:** monocular depth estimation, deep learning, computer vision, autonomous vehicle

## Abstract

It is important to estimate the exact depth from 2D images, and many studies have been conducted for a long period of time to solve depth estimation problems. Recently, as research on estimating depth from monocular camera images based on deep learning is progressing, research for estimating accurate depths using various techniques is being conducted. However, depth estimation from 2D images has been a problem in predicting the boundary between objects. In this paper, we aim to predict sophisticated depths by emphasizing the precise boundaries between objects. We propose a depth estimation network with encoder–decoder structures using the Laplacian pyramid and local planar guidance method. In the process of upsampling the learned features using the encoder, the purpose of this step is to obtain a clearer depth map by guiding a more sophisticated boundary of an object using the Laplacian pyramid and local planar guidance techniques. We train and test our models with KITTI and NYU Depth V2 datasets. The proposed network constructs a DNN using only convolution and uses the ConvNext networks as a backbone. A trained model shows the performance of the absolute relative error (Abs_rel) 0.054 and root mean square error (RMSE) 2.252 based on the KITTI dataset and absolute relative error (Abs_rel) 0.102 and root mean square error 0.355 based on the NYU Depth V2 dataset. On the state-of-the-art monocular depth estimation, our network performance shows the fifth-best performance based on the KITTI Eigen split and the eighth-best performance based on the NYU Depth V2.

## 1. Introduction

In the field of computer vision, depth estimation research is classically studied with various methods to obtain 3D information, and it is an important issue in fields such as visual Slam, 3D modeling, and autonomous driving. 

The sensor-based depth estimation method estimates depth by using Lidar and RGB-D cameras using light sensors. Recently, many research studies using lidar, which can grasp 3D information, have been conducted, forming the core of autonomous driving recognition technology. Lidar can measure distance with high accuracy by emitting light pulses to an object and detecting the reflected light via a sensor around the light source. However, in Lidar-based depth estimations, the higher the resolution, the lower the sharpness, and the higher the cost; moreover, the difference in the depth’s result depends on the internal and external parameters of each sensor.

As for the depth estimation method using the sensor, the left camera-based depth estimation has been classically studied based on the feature extraction of the camera input image via stereo matching the hand-crafted feature method. A stereo matching the estimated depth is performed by calculating the parallax between two images using an arbitrary point P in 3D space and epipolar geometry based on the p and p’ points projected on each left and right image. The stereo-matching method is greatly affected by the camera’s characteristics, and there is a problem in that the depth measured by lighting and phase changes is different. 

Since the development of deep learning, many studies have been conducted to solve the problem of depth estimations based on monocular cameras using techniques such as image segmentation and image regeneration. Deep-learning-based monocular camera depth estimation studies started to leap forward using the model [1] that applies CNN structure and end-to-end depth estimations at the pixel level using a deep neural network. Most recent state-of-the-art techniques employ an encoder that extracts feature points of an input image based on a deep convolutional neural network and a decoder that extracts depth information. Using a classification network that extracts the feature points of input images such as Resnet [2], Densenet [3], and ResNext [4], the backbone extracts dense features from the input three-channel RGB image and extracts the final depth map via the upsampling process based on the feature points extracted from the decoder. However, recently proposed depth estimation networks with good performance are fabricated based on a large transformer model with increased model depth. Therefore, the structure of the model becomes very large and difficult to use in real-time. In this study, we aim to construct a lightweight and high-performing model based on CNN without increasing the size of the model but by changing the decoder structure.

The contributions of this study are listed below:In this paper, we propose a lightweight supervised depth estimation network. The proposed network is based on generally used encoder–decoder depth estimation networks with the Laplacian image pyramid technique that emphasizes the boundaries between objects and the local planar guidance layer that guides the explicit relationship between features and final output. Figure 1 shows the accuracy of the depth prediction of the proposed network.

The proposed model was constructed via experiments by changing the backbone of the encoder and changing the structure of the decoder. The ConvNext [5]-Small model pre-trained on ImageNet-1K is used as a backbone network, and our model is composed entirely of 53 million parameters.The proposed model is not only lightweight but also shows high performance. It shows an absolute relative error of 0.054 on the KITTI Eigen split [6] and a root mean square error of 0.355 on the NYU Depth V2 dataset [7].

## 2. Related Works 

Deep-learning-based monocular camera depth estimation methods can be divided into supervised-learning-based depth estimation methods using point cloud data acquired from Lidar and RGB-D sensors and unsupervised learning depth estimation methods using stereo-matching techniques based on epipolar geometry. 

The supervised-learning-based depth estimation method estimates the final depth by calculating the difference between the input image and the ground truth image via a loss function using an image projected by point cloud data extracted from Lidar and RGB-D cameras. However, there are limitations in collecting depth ground truth data, and point cloud data collected using Lidar have a sparse value; various interpolation techniques are applied to overcome these problems. In the above process, additional work is required, and the data value acquired for each Lidar sensor is different, so additional parameter adjustments are required. In order to overcome this problem, an unsupervised depth estimation method has been proposed. 

The unsupervised depth estimation method is based on the stereo matching technique that simulates the human visual system as an input, the left image is the input, and the reconstruction loss is calculated based on the left disparity and right disparity obtained by learning the CNN network. The right image depth is estimated by regenerating and left image. The above method has the advantage of not requiring point cloud ground truth data because it estimates the depth based on epipolar geometry. 

However, the above method also has a disadvantage in that a complex network structure and various types of loss functions must be used, and solving the scale ambiguity problem is difficult. 

### 2.1. Supervised Depth Estimation

Supervised depth estimation is a method of extracting 3D depth information from 2D images using point cloud ground truth data acquired from 3D Lidar and RGB-D cameras. Eigen et al. [1] proposed a learning and depth map restoration method for depth information by constructing a multi-scale deep neural network with an encoder–decoder structure within a single network. Laina et al. [8] proposed a single architecture structure of the end-to-end method using a full convolution network of ResNet-based encoder–decoder structure and Huber loss. Fu et al. [9] approached the depth estimation problem as an ordinal regression problem and adopted a network structure that connects the discretization technique and multi-scale method in parallel using a spacing-increasing discretization strategy. Alhashim et al. [10] improved the accuracy and quality of depth maps by using transfer learning. 

The depth map generated via the deep learning network is trained based on the difference between the actual image and the generated depth map via loss functions such as route mean square error and silog error. The datasets used for learning and evaluation include the KITTI dataset, which consists of a Lidar-based outdoor dataset, and the NYU dataset, which consists of an indoor dataset based on RGB-D sensors.

Although there have been many advances in monocular camera-based depth estimation techniques with the introduction of deep neural networks, location information and spatial information are lost in the learning process, and it is difficult to estimate the exact boundary between objects in the upsampled process. 

In order to solve this problem, the fully connected layer used in the existing classification network is changed to a fully convolutional layer or U-Net [11] that connects the heat map extracted in the upsampling process by extracting features from each convolution layer of the backbone network and the ASPP [12] technique, which extended the field of view of the input feature more widely by applying dilated convolution.

BTS [13] provides an extended field of view in the encoding process by using the ASPP module in the existing encoder–decoder structure network and proposes a local planar guidance layer technique to guide the direct and explicit relationship between features and depth maps by applying the ray plane intersection formula to find the intersection of the camera ray and the 3D plane. Equation (1) is the ray plane intersection formula, where $n=\left(n_{1},n_{2},n_{3},n_{4}\right)$ the estimated plane coefficients, and (*ui, vi*) is the *k × k* patch-wise normalized coordinates of pixel *i*.
(1)$${\tilde{c}}_{i}=\frac{n_{4}}{n_{1}u_{i}+n_{2}v_{i}+n_{3}}$$

In the decoding process, the feature map that passed through the local planar guidance layer is shown in Figure 2, and it is reconstructed into four single vectors, experienced normalization, and sigmoid functions finally formed a map with the same size as the original image. By using the local planar guidance layer, it is possible to reduce the amount of computation and, at the same time, use four parameters to effectively restore the existing resolution. 

Lapdepth [14] applied the Laplacian image pyramid technique in the decoding process to improve performances and extract a clearer depth map. The ASPP module is applied to the encoding process to provide a wide field of view, and by linking the Laplacian residual RGB image to the extracted multi-scale feature map, it provides guidance to the location information and spatial information lost in the encoding process; the boundaries of the objects can be clearly distinguished. In addition, at each decoding step, the depth map is extracted and connected to the next decoding stage in the upsampling process to connect low-resolution depth information and high-resolution depth information. By combining local information at low resolutions and global information at high resolutions, a clear depth map can be extracted.

This paper proposes an effective and high-performance decoder by transforming the Laplacian image pyramid and local planar guidance layer. BTS [13] and Lapdepth [14] are our main competitive models.

### 2.2. Unsupervised Depth Estimation

The depth estimation method based on unsupervised learning obtains the final depth via the disparity between the left image and the right image based on the stereo-matching technique that simulates the human visual system. Zhou et al. [15] first proposed an unsupervised learning-based depth estimation technique for estimating video-based motion and depth estimations. Godard et al. [16] used the loss that regenerates the left image from the right image by calculating the reconstruction loss based on the left disparity and the right disparity obtained by passing the left input image via the CNN network and the loss that regenerates the right image from the left image.

Finally, a method for estimating the depth was proposed by applying the image regeneration method proposed by generative adversarial networks. Godard et al. [17] proposed an accurate and simple depth estimation network structure using the auto-mask technique based on the difference between the two images. Almalioglu et al. [18] proposed a network structure for estimating depth estimation and visual odometry using generative adversarial networks, and Wang et al. [19] proposed a network structure with improved performances via a recurrent neural network structure using continuous images.

### 2.3. ConvNext

ConvNext [5] is based on the vision transformer structure and ResNet50 model, which has recently shown state-of-the-art performances in various deep learning fields. It shows excellent performance as a model composed only of a convolutional neural network by changing the convolution method, activation function, and training method. The(1:1:3:1) ratio block configuration used in the former model was adopted, the block was changed to (3,3,9,3), and the grouped convolution proposed in ResNext [4] was used. The memory efficiency and accuracy improved by changing the bottleneck structure. Similar to the 7 × 7 size window used in the transformer, the existing 3 × 3 kernel was replaced with a 7 × 7 kernel, showing high accuracy. In addition, the performance improved by applying layer normalization instead of batch normalization and the pre-activation technique that proceeds from normalization -> activation function -> convolution in the configuration of convolution -> normalization -> activation function. Because the transformer model has little inductive bias, it has better performance than general convolution when learning using a large dataset. However, the ConvNext [5] model outperformed the transformer even on large datasets such as the ImageNet-22K model using only the inductive bias of the convolution. In this paper, the proposed network uses the ConvNext [5]-Small and Tiny models as the backbone and constructs the convolution layer using the layer structure of ConvNext [5].

## 3. Proposed Method

In this paper, as shown in Figure 3, we propose a modification of the local planar guidance layer method proposed by the existing monocular camera depth estimation network BTS [13] and a monocular camera depth estimation network via a modification of the Laplacian image pyramid method. The purpose of this study is to design a lightweight model that extracts accurate and clear depth maps using network designs that apply the latest deep learning techniques and various depth estimation techniques to an encoder–decoder-based monocular camera depth estimation network. In order to make a lighter and simpler model, the network was constructed using only convolution without adopting the transformer structure, which has been widely used in various deep learning methods recently. The network was composed of only CNN and was constructed by changing the detailed parameters and decoder structure.

### 3.1. Network Structure Details

In this paper, the network is constructed with an encoder–decoder structure and the ConvNext-Small model, which shows CNN-based state-of-the-art performance in the image classification field. Figure 3 shows the proposed network based on the Lapdepth structure. The encoder is divided into a total of 4 layers, and the skip-connection technique that extracts a feature map for each layer and connects it with each layer of the decoder is applied. The feature map, which is finally reduced to 1/16 resolution compared to the input image via the encoder, goes through the ASPP module that searches a wider receptive field with a small amount of computation. The feature map extracted through the ASPP module is upsampled through two processes in the decoding process. One feature map is upscaled to the original size via the local planar guidance layer and then down-sampled to 1/8 of the original size. Another feature map is up-sampled with the convolution process, and then the map is extracted from the 3rd layer of the encoder to compose one feature map. The feature map is downsampled through the configured feature map, and the local planar guidance layer is downsampled to 1/8 of the size of the original image inspired by the Laplacian pyramid technique; it is then downsampled to 1/16 of the size of the original image and then again at 1/8 of the size. It is concatenated with the image obtained via the difference with the image upsampled.

The new feature map generated by this process again passes through the local planar guidance layer and the upconvolution layer to generate the feature map and consists of a network that proceeds through a total of three iterative operations.

In the last decoding layer, the process of concatenating the feature points extracted for each layer is omitted, and the image is obtained by the difference from the image that is downscaled by 1/2 from the original image and then upscaled again. The up-convolved feature map and feature map passing through the local planar guidance layer was added, and then a dense depth map was extracted from the final convolution layer.

All convolution blocks were constructed using the pre-activation technique shown in Figure 4. The pre-activation technique was applied in the order of activation function -> normalization -> convolution in the convolution block composed of convolution -> normalization -> activation function used in the existing convolution block. The convolution block of the bottleneck structure is a technique that combines the feature map’s output from the convolution block and the input feature map for identity mapping when the feature map passes through the convolution block.

However, since additional functions are added via the activation function in each layer of the convolution block, an additional function is added between the input feature map and the output feature map. When the pre-activation technique is applied, only the convolutional feature map is output when passing through the convolution block because the feature map passes through the activation function before passing through the convolution block, and there is an improvement in performance in the resulting effect.

The layer that extracts the last depth map consists of two 7 × 7 convolution -> one 1 × 1 convolution -> a sigmoid function for extracting the final depth map using a small number of activation functions, and by arranging a 7 × 7 kernel-sized convolution, it is configured to search a wider area.

GELU was used as the activation function in all layers except for the last layer from which the depth map was extracted, and the network was constructed by applying the layer normalization technique. In order to attain a wider receptive field, all up-convolution layers except the ASPP Module and channel reduction layer are configured with 7 × 7 kernel size convolution.

Algorithm 1 describes the details of the proposed decoder structure using a pseudo code. The variable ei denotes a feature map extracted for each layer of the encoder that is skip-connected in the decoding process. Di  is a depth map generated by passing local planar guidance.
**Algorithm 1** The proposed decoder architecture.1. Input: Encoded feature at each encoder layer ei 2. Output: Estimated depth map Dfinal 3. Laplacian pyramid image: pi*4.* Decoder start
*5.   f*
=ASPP (e4) 6.   for (*i* = 4; *i* ≥ 1; *i* = *i* − 1)a.      Di =Local planar guidance (f)*b.    *
 f=Upconvolution (f) 7.     if *i* > 1 then
*a.      *
$\;f=concatenation\;(D_{i\;},\;\;p_{i},\;\;e_{i-1},\;\;f)$ 8.     **if** *i* == 1 then a.       $f=concatenation\;(D_{i\;},\;\;p_{i},\;\;f)$ 9.   end 10.   $D_{final\;}$ = $convolution^{7x7}\left(convolution^{7x7}\left(convolution^{1x1}\left(Sigmoid\left(f\right)\right)\right)\right)\times d_{max}$ 11.   return $D_{final\;}$ 12. Decoder end

### 3.2. Training Loss

The loss function used in the experiment was the scale-invariant logarithmic (SILog) loss presented by Eigen et al [1]. The scale invariant error is defined by the following equation.
(2)$$d_{i}=log_{y_{i}}-log_{\overset{´}{y_{i}}}$$

As in the equation in (2the pixel (*N*) containing valid depth information among the ground truth images in the equation below is obtained by calculating the difference by applying the log to the predicted depth map ($y_{i}$) and the ground truth ($y_{i}^{\prime }$).
(3)$$D(y_{i},\overset{´}{\;y_{i})=\frac{1}{N}\sum_{i}d_{i}^{2}-\frac{\lambda }{N^{2}}{\left(\sum_{i}di\right)}^{2}}$$

Moreover, λ = 0.85, and the calculated *D* is finally calculated as a value defined as α = 10, and a loss calculation is performed via the root calculation.
(4)$$L=\alpha \sqrt{D\left(y_{i,\;}\;{\overset{´}{y}}_{i}\right)}$$

## 4. Experiments and Evaluation

In this paper, experiments and evaluations were conducted for indoor and outdoor scenes using the KITTI Eigen split and NYU Depth V2 datasets.

### 4.1. KITTI Datasets

The KITTI dataset was acquired using Lidar, GPS, IMU, and a camera in outdoor scenes. It consists of a total of 61 outdoor scenes in four categories: ‘city,’ ‘residential,’ ‘road,’ and ‘campuses,’ and consists of a total of 24,185 RBG image data and depth ground truth data. It contains left and right stereo images with a resolution of 368 × 1244: 23,488 images from a total of 32 scenes were used for learning, and 697 images from 29 scenes were used for evaluation and testing.

### 4.2. NYU Depth V2 Dataset

The NYU Depth V2 [10] dataset consisted of a total of 464 internal scenes and was collected using a Kinect sensor. The dataset used for training consists of 24,231 images of a total of 249 scenes, and the dataset used for testing consists of 654 images of a total of 215 scenes used for testing. The resolution is 480 × 640 and contains RGB images and a depth map corresponding to each RGB image.

### 4.3. Experiment Environment and Details

For backbone networks, the ConvNext-small model used in this paper had weights pre-trained with ImageNet-1K, and AdamW was used as the optimizer. A total of 50 epochs were set. The batch size was set to 16, and the initial learning rate was set and finally decreased. The experiment was conducted with Pytorch version 1.10.1 in a Python 3.8 environment, and all experimental environments were performed using a device equipped with 4 NVIDIA Geforce GTX 1080ti. Detailed equipment information is shown in Table 1. Using accurate performance comparisons with Lapdepth and BTS methods used as the baseline, the ResNext-101 model was used as backbones in the network to make comparisons via training and performance evaluation. In the NYU Depth V2 dataset, RGB cameras and RGB-D sensors have different frame rates, so they cannot be set at the same time. The data set preprocessing method of [1] was followed. The depth image taken at the closest time based on each RGB image was used as the ground truth, and the one-to-many mapped RGB image was removed. The part where the depth value was missing from the selected depth map was processed as a mask. In addition, areas such as light reflection, doorways, and windows were filled with maximum or minimum depth information; the maximum depth value was set to 10 m, and the minimum depth value was set to 0.001. During training, a random crop method was applied to an existing 640 × 480 image with a resolution of 416 × 544, and for evaluation, the center crop method was applied within a range: width within (43, 608) and height within (45, 472). For the KITTI dataset, a random crop technique was applied with a resolution of 352 × 704 during training used in [20] was applied.

For each dataset, random color brightness, contrast adjustment, and horizontal flipping techniques were applied with a probability of 50% during training, and random rotations of [−1, 1] and [−2.5, 2.5] were used to prevent overfitting.

### 4.4. Evaluation

Performance evaluations were conducted using Abs_Rel, RMSE, Sq Rel, and Threshold performance evaluation indicators are shown in Equations (5)–(8) below. Two datasets, KITTI Eigen split and NYU Depth V2 were used.
(5)$$\mathrm{Abs}\;\mathrm{Rel}=\frac{1}{\left|T\right|}\sum_{\hat{d}\in T}\frac{\left|\tilde{d}-d\right|}{d}$$
(6)$$\mathrm{RMSE}\;=\sqrt{\frac{1}{\left|\;T\;\right|}\sum_{\hat{d}\in T}\frac{{||\tilde{d}-d||}^{2}}{d}}$$
(7)$$\mathrm{Sq}\;\mathrm{Rel}=\frac{1}{\left|\;T\;\right|}\sum_{\hat{d}\in T}\frac{{||\tilde{d}-d||}^{2}}{d}$$
(8)$$\mathrm{Threshold}\;=\%\;of\;{\tilde{d}}_{i}\;s.t.max\left(\frac{{\tilde{d}}_{i}}{d_{i}},\;\frac{d_{i}}{{\tilde{d}}_{i}}\right)=\delta <thr,$$

We proceeded using the same evaluation method on the NYU Depth V2 dataset as BTS [13] and Lapdepth [14], and the proposed network shows better performance, as shown in Table 2. As can be seen from the evaluation result in Table 2, the proposed network, which is applied by modifying the technique proposed by the two existing networks, shows better performance on the overall evaluation index.

Figure 5, comparing the depth maps extracted from the proposed networks with state-of-the-art methods. As can be seen in Figure 5, for outdoor environments, the proposed network accurately estimates the depth even for small objects on the road or a bicycle right next to a tree, and it can accurately distinguish a boundary between objects. Moreover, when comparing with the depth map extracted from the NeWCRFs model located at the top of the state-of-the-art techniques, a more sophisticated depth map is extracted compared to other networks. However, the KITTI dataset depth ground truth does not have a depth value for the upper part outside the lidar detection range. Therefore, in the learning process, the upper part of the image does not train well, so as can be seen in the other network results, our network also has limitations in that the depth of the upper part cannot be accurately measured in the inference process.

For a more accurate comparison, in Figure 6, color was applied to the depth map, and a detailed comparison with BTS and Lapdepth was performed via magnification. As shown in Figure 6, it can be confirmed that the proposed network accurately grasps the boundary between objects, such as people overlapping with the surrounding environment and traffic signs overlapping with surroundings, compared to BTS and Lapdepth, and it also more precisely grasps the shape of an object.

In Figure 7, as shown in the third and fourth pictures, the proposed network predicted depths more accurately in the curved part, such as the inside of the bathtub, between the trash can and the object on the bathtub, and distinguishing the boundary between a wall and a mirror on the same plane and predicting the depth accurately.

As shown in Table 3, The proposed model shows high performances and lower performances in total parameters not only in the convolution-based network but also in comparison with the transformer-based networks such as Adabins [21] and DPT-Hybrid [22]. Compared to Lapdepth, the number of parameters of the proposed network is about 71%, but the performance improved by 8.1%. Compared to BTS, the number of parameters of the proposed network increased by 12%, but the performance improved by 11%. Although the performance of the proposed model is not a state-of-the-art result compared with the model that uses a transformer backbone pre-trained with ImageNet-22k dataset, the proposed model has much better performances compared to the size of the model.

Table 4 shows the comparison results with the latest State-of-the-art Network after evaluation using the KITTI Eigen split Dataset. When ConvNext-Small was used as a backbone, the top 5 scores were recorded based on State-of-the-art papers.

Table 5 shows the results of the comparison with the latest state-of-the-art network after evaluations using the NYU Depth V2 dataset. When ConvNext-Small was used as a backbone, the top six scores were recorded based on the state-of-the-art paper.

As shown in Table 5 and Table 6, the proposed CNN-based model shows good performance compared to state-of-the-art networks and reaches the latest transformer-based models.

As shown in Table 6, a Performance comparison with BTS and Lapdepth was performed using the same backbone network ResNext-101 for the accurate performance evaluation of the proposed network. For the Same backbone networks, our network shows higher performance than the others. When ConvNext-small was used as a backbone network, Performance improved by more than 10% compared to Lapdepth and BTS.

### 4.5. Ablation Study

Figure 8 shows the change in the design of the decoder via a change in the synthesis method of the depth map extracted for each decoder layer in the process of designing the final layer of the decoder. Table 7 shows a comparison of the results of the KITTI Eigen split validation dataset with different decoder structures. The method then (a) summates the depth map extracted from all layers in the decoding process, and the other method involves (b) concatenating the depth map extracted from all LPG layers in the last layer. Method (C) is proposed in this.

As a result of the experiment, a sharper depth map was extracted by method (a), which summed all the depth maps extracted from each layer. However, in the process of adding images, substantial depth information is lost, and the performance deteriorates. When comparing the (b) method in which all depth maps extracted from each layer are concatenated at the last stage and (c) the method of concatenating only the D1 depth map in the last stage, the proposed (c) decoder structure has better performance.

Table 8 shows the performance evaluation results according to the backbone network change in the KITTI Eigen split. The proposed network is a model constructed using only convolution without using attention and a transformer, and in order to reduce the complexity and lightweight properties of the model, performance evaluations were performed using ConvNext-Tiny and ConvNext-Small models as backbone networks. Based on the KITTI Eigen split dataset, when ConvNext-Tiny and ConvNext-Small models were used as backbone networks, respectively, absolute relative errors at 0.057 and 0.054 were obtained. When the ConvNext-Small model was used as a backbone network, the highest performance was shown, and it can be confirmed that there is not much performance difference compared to the SOTA networks using other very large pretrained transformer models. In addition, the number of parameters based on the KITTI Eigen split dataset ConvNext-small Backbone is 53 M, and the number of parameters based on the ConvNext-Tiny model is 31 M, making it a very lightweight model compared to the existing transformer-based depth estimation network.

Table 9 shows the evaluation result of different Laplacian Pyramid methods. The Laplacian-based decoder proposed by Lapdepth [7]. In our network, it changed to filtering from the upsampled image to the original image, and the same effect as the low pass filter is applied to improve the performance. Table 10 is the KITTI Eigen split Evaluation result according to the upconvolution block change. Experiments were conducted using the original convolution block, pre-activation with 3 × 3 convolution, and pre-activation with 7 × 7 convolution, and the performance is best when the pre-activation with 7 × 7 convolution structure is adopted.

## 5. Conclusions

In this paper, we proposed a supervised monocular depth estimation network using various deep-learning techniques. We propose a simple and lightweight network that shows the performance of the state-of-the-art methods via the fusion and transformation of the local planar guidance layer and the Laplacian image pyramid technique using only convolution.

Via experiments using various deep learning techniques, such as large kernel sizes and pre-activation, the performance of the decoder-based depth estimation networks is beyond the performance shown by the existing CNN-based depth estimation networks.

Moreover, as can be seen from the generated depth map, a higher level of depth is predicted by accurately grasping the boundary of objects, and it shows good performance in both close and far predictions.

The next study will aim to make the model lighter and use TensorRT to configure a more accurate network that can guarantee real-time performance within an embedded board such as NVIDIA Xavier.

## Figures and Tables

**Figure 1 sensors-23-00845-f001:**
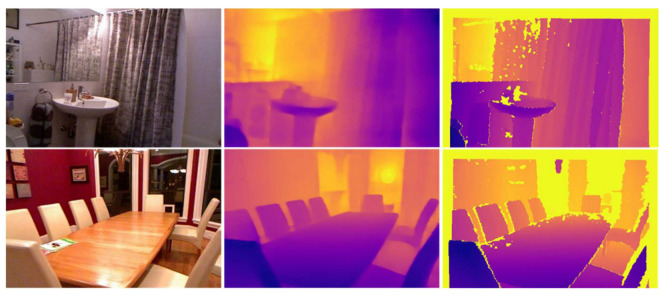
Example of an estimated depth map from the proposed network. From left to right: input image, our estimated depth map, and ground truth.

**Figure 2 sensors-23-00845-f002:**
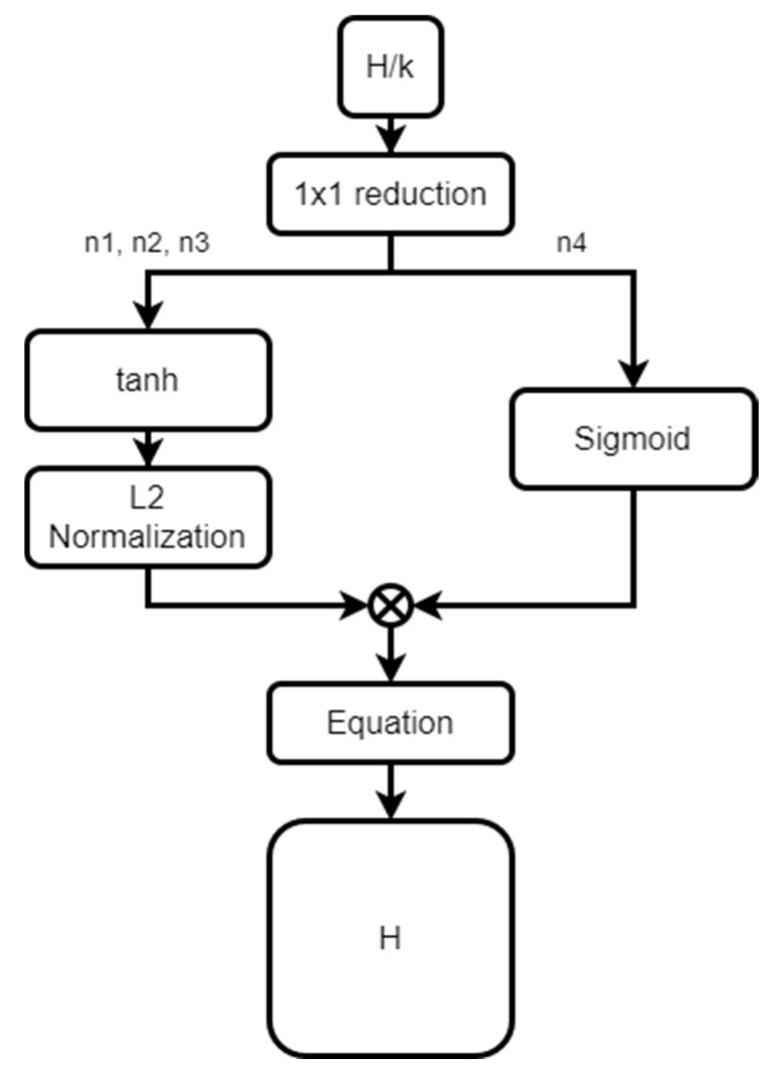
Local planar guidance.

**Figure 3 sensors-23-00845-f003:**
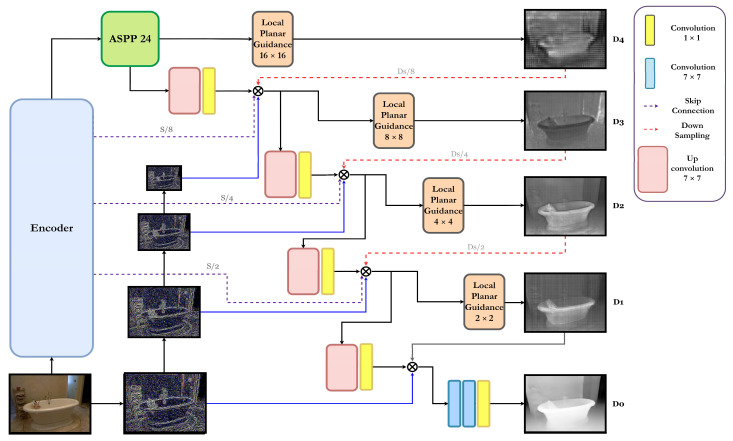
The proposed monocular depth estimation network architecture.

**Figure 4 sensors-23-00845-f004:**
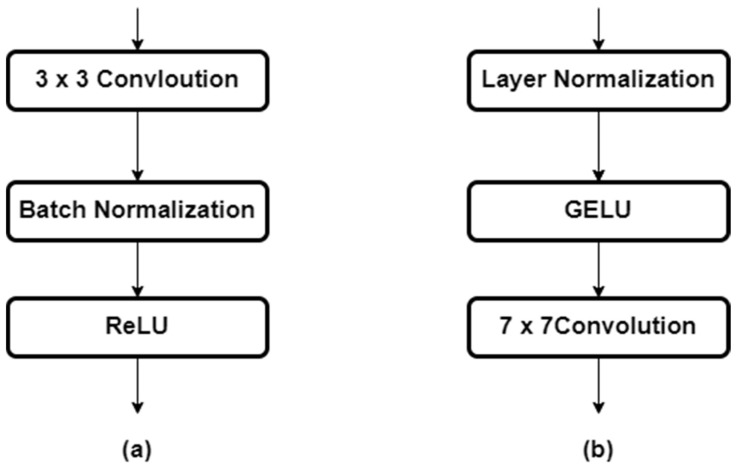
Typical convolution block (**a**) and pre-activation convolution block with layer normalization with 7 × 7 convolution (**b**).

**Figure 5 sensors-23-00845-f005:**
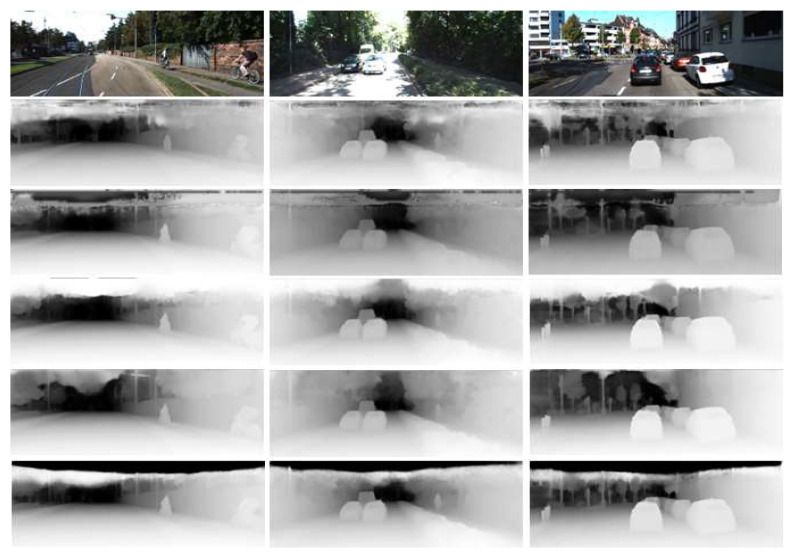
Result of predicted depth map on the KITTI Eigen split. From top to bottom: input image, BTS, Lapdepth, AdaBins, NeWCRFs, and ours.

**Figure 6 sensors-23-00845-f006:**
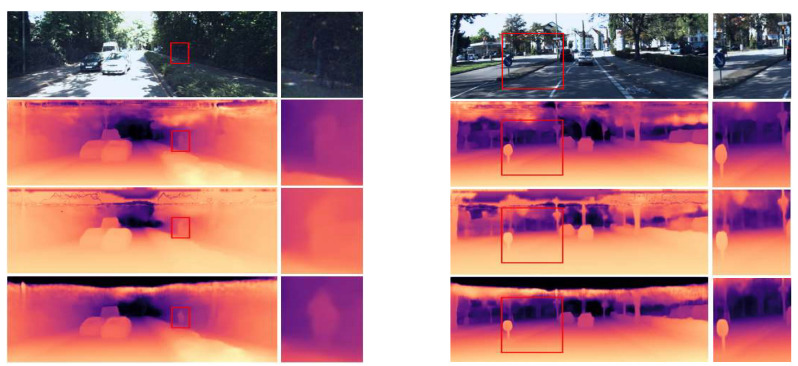
Predicted depth map on the KITTI Eigen split. Detailed comparison from top to bottom: original RGB, BTS, Lapdepth, and ours.

**Figure 7 sensors-23-00845-f007:**
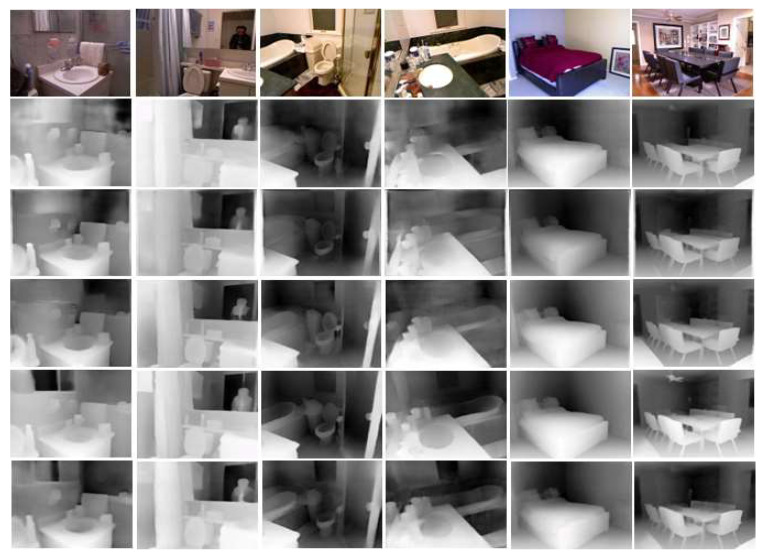
NYU Depth V2 depth map comparison from top to bottom: original RGB, BTS, Lapdepth, Adabins, NeWCRFs, and ours.

**Figure 8 sensors-23-00845-f008:**
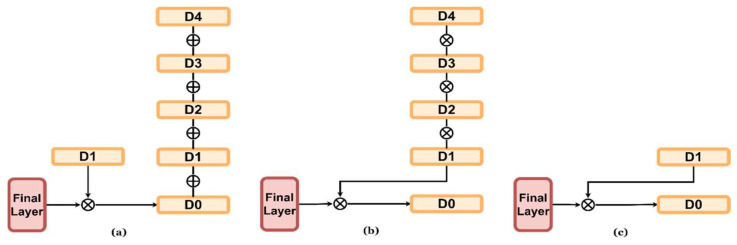
Various structures according to the change of the last layer of the decoder. From left to right: (**a**) add all estimated depth maps with the estimated depth map. (**b**) Concatenate all depth maps with the final layer. (**c**) Concatenate D1 with the final layer (proposed).

**Table 1 sensors-23-00845-t001:** Experiment with PC specifications.

CPU	Intel I9 7920X
RAM	128 GB
OS	Ububtu 18.04LTS
GPU	NVIDIA GeForce GTX 1080ti × 4

**Table 2 sensors-23-00845-t002:** Performance comparison using NYU Depth V2 dataset with BTS and Lapdepth.

Method	Lower Is Better			Higher Is Better		
	RMSE	Abs_Rel	Sq_Rel	$\delta_{1}$	$\delta_{2}$	$\;\;\;\delta_{3}$
BTS [13]	0.392	0.110	0.066	0.885	0.978	0.995
Lapdepth [14]	0.384	0.105	-	0.895	0.983	0.996
**Ours**	**0.355**	**0.102**	**0.053**	**0.904**	**0.985**	**0.998**

**Table 3 sensors-23-00845-t003:** Performance comparison results and network parameters using NYU Depth V2 dataset.

Method	Parms	Lower Is Better			Higher Is Better		
		RMSE	Abs_Rel	Sq_Rel	$\delta_{1}$	$\delta_{2}$	$\;\;\;\delta_{3}$
BTS [13]	**47 M**	0.392	0.110	0.066	0.885	0.978	0.995
Lapdepth [14]	74 M	0.384	0.105	-	0.895	0.983	0.996
Adabins [21]	78 M	0.364	0.103	-	0.899	0.984	0.997
DPT-Hybrid [22]	123 M	0.357	0.110	-	0.904	0.988	0.994
NeWCRFs [23]	270 M	**0.334**	**0.095**	**0.045**	**0.922**	**0.992**	**0.998**
**Ours**	53 M	0.355	0.102	0.053	0.904	0.985	0.998

**Table 4 sensors-23-00845-t004:** Performance comparison using the KITTI Eigen split with the state-of-the-art networks.

Method	Cap	Lower Is Better			Higher Is Better		
		Abs_Rel	RMSE	Sq_Rel	$\delta_{1}$	$\delta_{2}$	$\delta_{3}$
DORN [9]	0–80 m	0.072	2.727	-	0.932	0.984	0.994
DPT-Hybrid [22]	0–80 m	0.062	2.573	-	0.959	0.995	0.999
BTS [13]	0–80 m	0.060	2.798	0.241	0.955	0.993	0.998
Lapdepth [14]	0–80 m	0.059	2.446	0.212	0.962	0.994	0.999
Adabins [21]	0–80 m	0.058	2.360	0.190	0.964	0.995	0.999
GLPDepth [24]	0–80 m	0.057	2.297	-	0.967	0.996	0.999
**Ours**	0–80 m	0.054	2.252	0.179	0.969	0.996	0.999
MonoDelsnet [25]	0–80 m	0.053	2.101	0.161	0.969	0.996	0.999
DepthFormer [26]	0–80 m	0.052	2.143	0.158	**0.975**	0.997	0.999
NeWCRFs [23]	0–80 m	0.052	2.129	0.155	0.974	0.997	0.999
BinsFormer [27]	0–80 m	**0.052**	**2.098**	**0.151**	0.974	**0.997**	0.999

**Table 5 sensors-23-00845-t005:** Performance comparison using the NYU Depth v2 Dataset with state-of-the-art networks.

Method	Lower Is Better			Higher Is Better		
	RMSE	Abs_Rel	Sq_Rel	$\delta_{1}$	$\delta_{2}$	$\delta_{3}$
BTS [13]	0.392	0.110	0.066	0.885	0.978	0.995
Lapdepth [14]	0.384	0.105	-	0.895	0.983	0.996
Adabins [21]	0.364	0.103	-	0.899	0.984	0.997
DPT-Hybrid [22]	0.357	0.110	-	0.904	0.988	0.994
P3Depth [28]	0.356	0.104	-	0.898	0.981	0.996
**Ours**	0.355	0.102	0.053	0.904	0.984	0.998
LocalBins [29]	0.351	0.098	-	0.91	0.986	0.997
GLPDepth [24]	0.345	0.100	-	0.915	0.988	0.997
Depthformer [26]	0.339	0.096	-	0.921	0.989	0.998
NeWCRFs [23]	0.334	0.095	0.045	0.922	**0.992**	**0.998**
Binsformer [27]	**0.330**	**0.094**	-	**0.925**	0.989	0.997

**Table 6 sensors-23-00845-t006:** KITTI Eigen split performance comparison with BTS and Lapdepth using the same backbone network.

Method	Backbone	Parms	Lower Is Better	
			RMSE	Abs_Rel
BTS [13]	ResNext-101	113 M	2.798	0.060
Lapdepth [14]	ResNext-101	74 M	2.446	0.059
**Ours**	ResNext-101	84 M	2.389	0.057
	**ConvNext-Small**	**53 M**	**2.252**	**0.054**

**Table 7 sensors-23-00845-t007:** KITTI Eigen split validation results according to decoder structure. (a) Sum of all depth maps extracted from each layer. (b) Concatenation of all depth maps extracted from each layer. (c) Concatenate D1 with the final layer (proposed).

Method	Lower Is Better	
	RMSE	Abs_Rel
(a)	2.337	0.057
(b)	2.322	0.056
**(c) (proposed)**	**2.252**	**0.054**

**Table 8 sensors-23-00845-t008:** KITTI Eigen split evaluation result, with different backbone networks (except for the backbone network, other environments are the same).

Backbone	Parms	Lower Is Better			Higher Is Better		
		RMSE	Abs_Rel	Sq_Rel	$\delta_{1}$	$\delta_{2}$	$\;\;\;\delta_{3}$
ResNext-101	84 M	2.389	0.057	0.198	0.963	0.994	0.999
ConvNext-Tiny	**31 M**	2.337	0.057	0.191	0.965	0.995	0.999
**ConvNext-Small**	53 M	**2.252**	**0.054**	**0.179**	**0.969**	**0.996**	**0.999**

**Table 9 sensors-23-00845-t009:** KITTI Eigen split evaluation results with different Laplacian Pyramid methods (except the Laplacian Pyramid, other environments are the same).

Method	Lower Is Better			Higher Is Better		
	RMSE	Abs_Rel	Sq_Rel	$\delta_{1}$	$\delta_{2}$	$\delta_{3}$
Laplacian Pyramid	2.301	0.055	0.182	0.967	0.996	0.999
**Inversed Laplacian** **Pyramid(proposed)**	**2.252**	**0.054**	**0.179**	**0.969**	**0.996**	**0.999**

**Table 10 sensors-23-00845-t010:** KITTI Eigen split Evaluation Result, With Different upconvolution blocks (Except upconvolution block, other environments are same).

Method	Lower Is Better			Higher Is Better		
	RMSE	Abs_Rel	Sq_Rel	$\delta_{1}$	$\delta_{2}$	$\delta_{3}$
Original	2.341	0.058	0.190	0.964	0.995	0.999
Pre-activation (3 × 3)	2.292	0.055	0.183	0.968	0.996	0.999
**Pre-activation (7** ** × 7)**	**2.252**	**0.054**	**0.179**	**0.969**	**0.996**	**0.999**

## Data Availability

Not applicable.

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
