# Peer review of "Monocular Depth Estimation Using a Laplacian Image Pyramid with Local Planar Guidance Layers"

_sensors, 2023, doi:10.3390/s23020845_

Round 1

Reviewer 1 Report

This paper proposes a depth estimation network of encoder-decoder structure using the Laplacian Pyramid and Local Planar Guidance method. The work has some reference significance, but this paper mainly has the following problems:

 1. In the introduction, the authors need to explain the innovation points to analyze the differences between the work in this paper and other works.

2. Formula (1) lacks the necessary explanation and explanation.

3. The evaluation of a method (item 3) could not be considered a contribution, even if outstanding results are obtained. Please revisit your contribution

4. Related work should be more than a simple pile of literature. Literature should be summarized and analyzed to explain the relationship between references and this paper.

5. In the method, the algorithm flow chart or pseudocode can be used to display the method in this paper better.

6. The experimental part needs to improve unnecessary hyperparameter ablation experiments.

7. Lack of necessary analysis and interpretation of visualized results.

8. The method proposed in this paper does not achieve SOTA. What is the significance and importance of the work in this paper?

Reviewer 2 Report

In photos taken outdoors, on the streets, you will notice that your algorithm blackens the upper parts of the photo. The depth map is black in these parts, which should mean that the objects in the photo are in the background, or very far away. This is not always true. I have not found an explanation. Here is a possibility of completion. 
